# Compensatory Growth and Physiological Protective Mechanisms of *Populus talassica* Kom. × *Populus euphratica* Oliv. in Response to Leaf Damage

**Meng-Xu Su [1,2], Zhan-Jiang Han [1,2,*], Ying Liu [1,2], Zhen Zhao [3] and Jia-Ju Wu [1,2]**

[1] College of Life Science and Technology, Tarim University, Alar 843300, China; sumengxu403@163.com (M.-X.S.); liuying836354822@126.com (Y.L.); wujiaju1115@163.com (J.-J.W.)
[2] Xinjiang Production and Construction Corps Key Laboratory of Protection and Utilization of Biological Resources in Tarim Basin, Alar 843300, China
[3] Institute of Agricultural Sciences, The First Division of Xinjiang Production and Construction Corps, Alar 843300, China; zhaozhen901016@163.com
* Correspondence: hanzhanjiang@taru.edu.cn

**Abstract:** The compensatory growth and defensive capabilities of woody plants after damage are crucial to their large-scale promotion and economic value. Here, *Populus talassica* × *Populus euphratica* were subjected to artificial defoliation treatments that simulated leaf damage [25% ($D_{25}$), 50% ($D_{50}$), and 75% ($D_{75}$) leaf removal] to study the growth, anatomical, and physiological characteristics. The results showed that $D_{25}$ and $D_{50}$ treatments significantly increased the growth parameters, such as leaf length, leaf area, and specific leaf area, but did not affect the distributions of root and stem biomasses compared with the CK. However, the $D_{75}$ treatment significantly decreased most growth parameters. The time required for the chlorophyll content to recover increased along with the damage intensity as follows: $D_{25}$, high-flat-high; $D_{50}$, low-high-flat; and $D_{75}$, low-flat-high. Furthermore, leaf damage significantly reduced stomatal density, whereas the stomatal width, area, opening, and *Pn* significantly increased by 8.59%, 8.40%, 23.27%, and 31.22%, respectively, under the $D_{50}$ treatment, generating a photosynthetic compensation response. The leaf anatomical parameters increased along with damage intensity, except spongy tissue thickness, which decreased, while the stem anatomical parameters showed trends of first increasing and then decreasing, reaching maxima under the $D_{50}$ treatment. The enzymes showed an increasing and then decreasing trend as the damage time increased. After 1 d of treatment, CAT, POD, and PAL activities peak at $D_{75}$, in contrast to a peak of SOD activity at $D_{50}$. Overall, these findings indicate that it is advisable to keep the amount of leaf damage within 50%. The leaf damage can have an impact on the growth of *P. talassica* × *P. euphratica*. They adjusted their resource allocation strategy and physiological defense capacity by increasing the chlorophyll content, improving photosynthetic capacity, changing stem and leaf anatomy, and increasing defense enzyme activity levels, thereby improving their damage tolerance and adaptability.

**Keywords:** *P. talassica* × *P. euphratica*; leaf damage; compensatory growth; defense enzymes; biomass allocation





## 1. Introduction

Plants often have to cope with complex and changing environmental challenges during long-term growth. Climate factors, animal feeding, and management measures can cause mechanical damage to plants, affecting their growth and development [1,2]. Studying the ability of plants to recover from mechanical damage is of great significance for their survival. Mechanical damage to woody plants often weakens tree growth, affecting plant growth physiology [3], biomass allocation [4], photosynthetic characteristics [5], and carbon sequestration capacity [6]. In severe cases, it slows lateral root or branch growth, which can even result in tree mortality [7,8].

For example, the same pest that causes defoliation can result in different mortality rates in plants [9]. The same leaf removal treatments all significantly reduce plant height and biomass; however, different genera have different tolerances [6]. Thus, plants have evolved various compensation strategies to resist damage stress, which allow them to repair and prevent future damage as well as increase their adaptability to environmental changes. These strategies include changing specific leaf area (SLA), increasing chlorophyll content, and enhancing photosynthetic rate [5,10], with the long-term maintenance of the photosynthetic compensation capacity being related to the amount of leaf loss and tree species [11,12]. In addition, leaf damage significantly increases the activities of defense enzymes, such as catalase (CAT), peroxidase (POD), and phenylalanine ammonia-lysae (PAL), in plants, in both damaged and undamaged leaves [13,14]. Consequently, exploring the compensatory growth and defense strategies of plants after damage will help increase our understanding of damage tolerance and provide a basis for plant management during recovery growth.

*Popular talassica* × *Popular euphratica*, excellent tree varieties, are the result of cross-breeding *P. talassica* as the female parent with *P. euphratica* as the male parent [15]. The tree is widely distributed in northwest China. It is a perennial deciduous tree that is tolerant to salinity, heat, and drought, and it provides high-quality seedlings for reforestation in saline, arid, and sandy areas, with broad promotional prospects in the construction of artificial forests for wind and sand control and increasing green areas. Our research group was previously engaged in research on the adaptation of *P. talassica* × *P. euphratica* to saline environments and artificial forests in southern Xinjiang, China. *P. talassica* × *P. euphratica* exhibit strong salt and drought tolerance and have high adaptability in the saline alkali regions of the Tarim Basin [16]. However, during the field investigation of the artificial cultivation of *P. talassica* × *P. euphratica* in Xinjiang, it was determined that its planting area and ecological functions were seriously affected by mechanical damage caused to seedlings by thinning and defoliation, animal gnawing, and wind and sand infestation, as well as the accompanying pest feeding. Therefore, the adaptability of this poplar to damage is related to its extension area, economic value, and ecological role.

In this study, two-year-old *P. talassica* × *P. euphratica* seedlings served as experimental materials to study growth and development, photosynthetic physiology, anatomical structure, and defense enzyme activities through different mechanical damage intensities. The regulatory mechanisms of the physiological defense capabilities of *P. talassica* × *P. euphratica* on their damage tolerance and compensatory growth were explored in order to clarify the adaptive strategies of *P. talassica* × *P. euphratica* under different mechanical damage intensities. The purpose of this study was to provide a new theoretical basis for the integrated management of *P. talassica* × *P. euphratica* plantations and further address the interactions between forest trees and the environment.

## 2. Materials and Methods

### 2.1. Overview of the Study Site

This study site was located in the *P. talassica* × *P. euphratica* nursery of the seedling base of the 10th Regiment of the 1st Division of the Xinjiang Production and Construction Corps, China (81°18′08″ E, 40°36′13″ N, and an altitude of 1014 m), and the research was carried out from May to August 2022. The area has an extreme continental, arid desert climate in a warm temperate zone. The average annual temperature is 12.1 °C; the average number of sunshine hours is 2568.5 h; and the average annual precipitation is 54.1 mm. There is sufficient light, large temperature differences between day and night, scarce annual rainfall, and strong surface evaporation. The soil type of the selected test sample plot is loam, and the basic physical and chemical properties are as follows: pH, 8.03; soil conductivity, 609.43 μS/cm; salt salinity, 2.87 g/kg; soil density, 1.55 g/cm$^3$; organic matter content, 21.34 g/kg; and the contents of alkali-hydrolyzed nitrogen, available potassium, and available phosphorus were 17.08 mg/kg, 143.07 mg/kg, and 22.05 mg/kg, respectively.

### 2.2. Plant Material and Experimental Treatments

The test materials were two-year-old *P. talassica* × *P. euphratica* seedlings, which were selected and divided into four groups of 50 plants each with good, consistent growth and disease-free leaves. Three of the groups were subjected to damage (cutting) treatments on a number of leaves in a bottom-up order: D25% ($D_{25}$): cutting one leaf per four leaves; D50% ($D_{50}$): cutting one leaf per two leaves and D75% ($D_{75}$): cutting three leaves; per four leaves. The damage consisted of cutting the whole leaf off with only the petiole remaining while avoiding damage to the terminal buds so that the branches could develop fully. Healthy plants with uncut leaves were used as the control group (CK), and all the seedlings were labeled and marked (Table 1).

**Table 1.** Leaf damage treatment of *Popular talassica* × *Popular euphratica*.

| Treatment | Damage Treatment |
|---|---|
| CK | no defoliation treatment |
| $D_{25}$ | cutting 1 leaf per 4 leaves |
| $D_{50}$ | cutting 1 leaf per 2 leaves |
| $D_{75}$ | cutting 3 leaves per 4 leaves |

### 2.3. Measurement of Growth Indexes

From each treatment, 10 plants were randomly selected for measurement. On the day before treatment (day 0), plant height, ground diameter, and crown width were measured, and they were measured again every 30 d. The differences between consecutive measurements, 0–30 d and 30–60 d after treatment, were used as the growth amounts for plant height, ground diameter, and crown width. Three plants were randomly selected from each treatment and taken to the laboratory at 60 d. After cleaning with deionized water, the samples were then dried in an electrothermal constant-temperature drying oven at 105 °C for 30 min and at 80 °C until a constant mass was attained. Then, the samples were weighed (accurately to 0.001 g). The compensation index is the ratio of the biomass of the treated plants to that of the control plants [17], with a compensation index greater than one being over-compensated, less than one being under-compensated, and equal to one being equal to compensation. The root-to-shoot ratio (R/S), seedling quality index (QI) [18], and compensation index (CI) [17] were calculated as follows:

$$\frac{R}{S} = \frac{UB}{AB} \tag{1}$$

$$QI = \frac{WB}{\frac{PH}{GD} + \frac{AB}{UB}} \tag{2}$$

$$CI = \frac{TB}{CB} \tag{3}$$

where UB represents under-ground biomass, AB represents above-ground biomass, WB represents whole plant biomass, PH represents plant height, GD represents ground diameter, TB represents treated plant biomass, and CB represents control plant biomass.

### 2.4. Measurement of Leaf Morphological Indexes

Healthy leaves were collected from the middle of the plant branches at the end of the experiment, and leaf length (LL), leaf width (LW), and single leaf area (LA) were measured using a portable leaf area meter LI-3000C (LI-COR Inc., Lincoln, Nebraska, USA). After weighing, the leaves were placed into an oven at 105 °C for 15 min and at 80 °C until

a constant mass was attained. They were then weighed (accurate to 0.001 g), providing the leaf dry weights (LDWs). The specific leaf area (SLA) was calculated as follows:

$$SLA = \frac{LA}{LDWs} \tag{4}$$

### 2.5. Measurements of Chlorophyll Content

Fresh leaves were collected at 5, 10, 30, and 60 d after treatment and extracted using 95% ethanol [19]. Briefly, 0.2 g of fresh leaves (avoiding the main leaf veins) were cut and placed in 50 mL of a 95% ethanol extraction solution. Using a UV-1601 UV-Visible spectrophotometer [Beijing Beifen-Ruili Analytical Instrument (Group) Co., Ltd., Beijing, China] at wavelengths of 649 nm and 665 nm, the contents of chlorophyll a (Chl a), chlorophyll b (Chl b), and total chlorophyll (Total Chl) were calculated as follows:

$$Chl\ a = \frac{(13.95A_{665} - 6.88A_{649}) \times V}{W \times 1000} \tag{5}$$

$$Total\ Chl = Chl\ a + Chl\ b \tag{6}$$

$$Chl\ b = \frac{(24.96A_{649} - 7.32A_{665}) \times V}{W \times 1000} \tag{7}$$

where A = optical density at 649 and 665 nm, V = final volume (mL), and W = leaf tissue fresh weight (g).

### 2.6. Measurements of Photosynthetic Parameters

The leaf photosynthetic parameters were measured from 9:00–12:00 AM in July 2022. Net photosynthetic rate (*Pn*), transpiration rate (*E*), intercellular $CO_2$ concentration values (*Ci*), and stomatal conductance (*Gs*) were measured on the 4th–6th healthy mature leaves of the middle branch. Using a LI-6400XT portable photosynthesizer (LI-COR Inc., Lincoln, Nebraska, USA) under a photosynthetically active radiation of 1200 $\mu mol/m^2/s$ (*PAR*), the temperature is 25 °C, and the atmospheric $CO_2$ concentration (*Ca*) is 390 $\mu mol/mol$. In total, five plants were measured per treatment, and five leaves from the same part of each plant were selected for measurements, which were averaged. Stomatal limitation values (*Ls*) and water-use efficiency (*WUE*) were calculated as follows:

$$Ls = 1 - \frac{Ci}{Ca} \tag{8}$$

$$WUE = \frac{Pn}{E} \tag{9}$$

### 2.7. Measurements of Stomatal Indexes

Leaf stomata were extracted 30 d after treatment using the nail polish blotting method [20]. Briefly, functional leaves from the same parts of each treated plant were taken on a sunny day between 10:00 and 11:00 AM. After removing the dust from the leaf surface with a skimmed cotton pad, clear, colorless nail polish was applied evenly to the leaf epidermis. After it had dried, transparent tape was placed on top and pressed to the surface. Then, the tape with the leaf stomata was carefully removed with forceps and glued to a slide. Leaf stomatal features were observed and photographed under a Nikon ECLIPSE Si light microscope (Nikon, Tokyo, Japan) equipped with a camera (OPLENIC CORP, Hangzhou, China). Stomatal length, width, area, density, and aperture (the widest part of the stomatal inner diameter) were measured using ImageJ analysis software (IJ 1.46r, National Institutes of Health, Bethesda, MD, USA) [20]. For each treatment, five mature leaves were selected from the same part of each plant, five different microscope fields were randomly selected, and five photos were taken per field. The

stomatal area index is the total area of stomata per unit leaf area. The stomatal shape index is a measure of the complexity of a single stomatal shape derived by calculating its deviation from a circle having the same area. When the stomata are circular, the shape index is 1. The flatter and longer the stomatal shape, the greater the stomatal shape index [21]. The proportion of opened stomata, stomatal shape index, and area index were calculated as follows:

$$\text{proportion of opened stomatal} = \frac{\text{number of opened stomatal per area}}{\text{number of stomatal per area}} \times 100\% \quad (10)$$

$$\text{stomatal shape index} = \frac{\text{SP}}{2 \times \sqrt{\pi \times \text{SA}}} \quad (11)$$

$$\text{stomatal area index} = \text{SD} \times \text{SA} \quad (12)$$

where SP is the stomatal perimeter, SA is the stomatal area, and SD is the stomatal density.

### 2.8. Measurements of Anatomical Indexes

Leaf and branch sections were made 30 d after treatment using the paraffin section method [22]. Six plants were chosen from each treatment, and leaves and branches were selected. Leaves were cut across the widest part (preserving the main veins), and the branches were cut into the 1-cm stem segments lacking leaf buds. Samples were immediately fixed in FAA solution (70% ethanol: glacial acetic acid: formaldehyde = 90:5:5) in the field for preservation. The anatomy was observed and photographed under a Nikon ECLIPSE Ci-L light microscope (Nikon Japan) with a Ds-Fi3 camera. Leaf indicators, such as leaf thickness, upper epidermal thickness, palisade tissue thickness, spongy tissue thickness, palisade/spongy tightness of leaf tissue structures, as well as epidermal thickness, cortical thickness, phloem thickness, xylem thickness, and pith diameters of stem segments, were measured using ImageJ analysis software. In total, five fields of view were observed per leaf and stem section, and 10 sets of data were measured for each indicator. The palisade/spongy and tightness of leaf tissue structures were calculated as follows [22]:

$$\text{palisade/spongy} = \frac{\text{palisade tissue thickness}}{\text{spongy tissue thickness}} \quad (13)$$

$$\text{tightness of leaf tissue structure} = \frac{\text{palisade tissue thickness}}{\text{leaf thickness}} \times 100\% \quad (14)$$

### 2.9. Measurements of Biochemical Indexes

Fresh leaves were collected at 1, 4, 7, and 10 d after treatment. The mixed leaves were quickly stored in an ultra-low-temperature refrigerator at $-80\ ^\circ\text{C}$ for use in biochemical assays. The CAT level was determined using the micro-method [23], and the absorbance of the reaction system was determined at 240 nm. Each gram of tissue catalyzes one unit per minute in the reaction system, and 1 µmol $H_2O_2$ degradation is defined as an enzyme activity unit. The levels of SOD were determined using the WST-1 method [24], and the absorbance of the reaction system was determined at 450 nm. The SOD enzyme activity in the reaction system was defined as one enzyme activity unit at 50% inhibition in the xanthine oxidase-coupled reaction system. The amount of POD was determined using guaiacol colorimetry [23], and the absorbance of the reaction system was determined at 470 nm. The A470 change of 0.005 per minute per gram of tissue in each 1-mL reaction system is defined as an enzyme activity unit. The amount of PAL was determined by trans-cinnamic acid colorimetry [25], and the absorbance of the reaction system was determined at 290 nm. The A290 change of 0.05 per minute per gram of tissue in each 1-mL reaction system is defined as an enzyme activity unit. All of the above were measured using an enzyme activity assay kit from Solarbio Science and Technology Biotechnology Co., Ltd.,

(Beijing, China). All the physiological index measurements were repeated three times, and the statistical results were averaged.

*2.10. Statistical Analysis*

Statistical analyses were conducted using SPSS 25.0 (IBM Corp., Armonk, NY, USA). The experimental data obtained were tested using a one-way ANOVA to examine the effects of different mechanical damage intensities on growth and development, photosynthetic physiology, anatomical structure, and defense enzyme activities. Each index was tested for significance with Duncan's method for multiple comparisons ($p < 0.05$ significance level). The data were expressed as means $\pm$ SE. Origin 2022 (OriginLab, Northampton, MA, USA) was used for creating graphs.

## 3. Result

*3.1. Damage Treatment Effects on Growth Indicators*

The effects of different damage intensities on the plant height growth, ground diameter growth, and crown width growth of *P. talassica* $\times$ *P. euphratica* are shown in Table 2. At 0–30 d, compared with CK, there was no significant change in the plant height growth under the $D_{25}$ and $D_{50}$ treatments. The maximum and minimum mean values of the ground diameter growth were observed under the $D_{25}$ and $D_{75}$ treatments, respectively. The crown width growth showed a decreasing trend, with negative growth under the $D_{75}$ treatment. At 30–60 d, the plant height growth was significantly greater in the $D_{25}$ treatment than in the CK and other treatment groups. The ground diameter growth inhibition was alleviated; therefore, the difference from the CK decreased compared to the growth of the 0–30 d. In addition, the crown width growth still exhibited growth inhibition.

**Table 2.** Effect of leaf damage on *P. talassica* $\times$ *P. euphratica* growth.

| Treatment | Plant Height Growth (cm) | | Ground Diameter Growth (cm) | | Crown Width Growth (cm) | |
|---|---|---|---|---|---|---|
| | 0–30 d | 30–60 d | 0–30 d | 30–60 d | 0–30 d | 30–60 d |
| CK | 12.180 ± 0.697 a | 3.400 ± 0.499 b | 0.969 ± 0.090 b | 0.971 ± 0.090 a | 15.460 ± 0.544 a | 9.680 ± 0.711 a |
| $D_{25}$ | 11.500 ± 1.114 a | 5.640 ± 0.898 a | 1.491 ± 0.070 a | 1.004 ± 0.070 a | 16.730 ± 0.892 a | 3.910 ± 0.859 b |
| $D_{50}$ | 11.080 ± 0.708 a | 2.800 ± 0.042 b | 1.105 ± 0.104 b | 0.974 ± 0.042 a | 8.700 ± 0.293 b | 4.520 ± 0.590 b |
| $D_{75}$ | 8.180 ± 0.491 b | 2.300 ± 0.257 b | 0.652 ± 0.035 c | 0.568 ± 0.061 b | −4.200 ± 0.110 c | 3.020 ± 0.200 b |

Note: The data in the table represent Mean ±standard errors, and different lowercase letters indicate significant differences ($p < 0.05$).

*3.2. Damage Treatment Effects on Leaf Growth Indicators*

The effects of different damage intensities on the leaf traits of *P. talassica* $\times$ *P. euphratica* are presented in Figure 1. The LL and LA showed trends of first increasing and then decreasing after damage treatments (Figure 1A,B). Compared with CK, the LL and LA increased significantly under the $D_{25}$ and $D_{50}$ treatments; however, the differences between the two treatments were not significant. However, the LW trend was opposite and significantly decreased under the $D_{50}$ treatment compared with CK (Figure 1C). The changing trend of SLA (Figure 1C) is similar to that of LA.

*3.3. Damage Treatment Effects on Biomass*

Changes in the *P. talassica* $\times$ *P. euphratica* biomass under the damage treatments are shown in Figure 2. Compared with CK, the root and stem biomasses were significantly lower under the $D_{75}$ treatment, by 38.85% and 13.75%, respectively. However, no significant changes were found under the $D_{25}$ and $D_{50}$ treatments (Figure 2A,B). Leaf biomass was significantly lower after the three damage treatments, $D_{25}$, $D_{50}$, and $D_{75}$, than in CK, with reductions of 22.73%, 31.97%, and 56.03%, respectively (Figure 2C). Compared with CK, total biomass was significantly lower after the damage treatments, by 8.99%, 12.37%, and

30.43%, respectively, with a non-significant difference between the $D_{25}$ and $D_{50}$ treatments (Figure 2D).

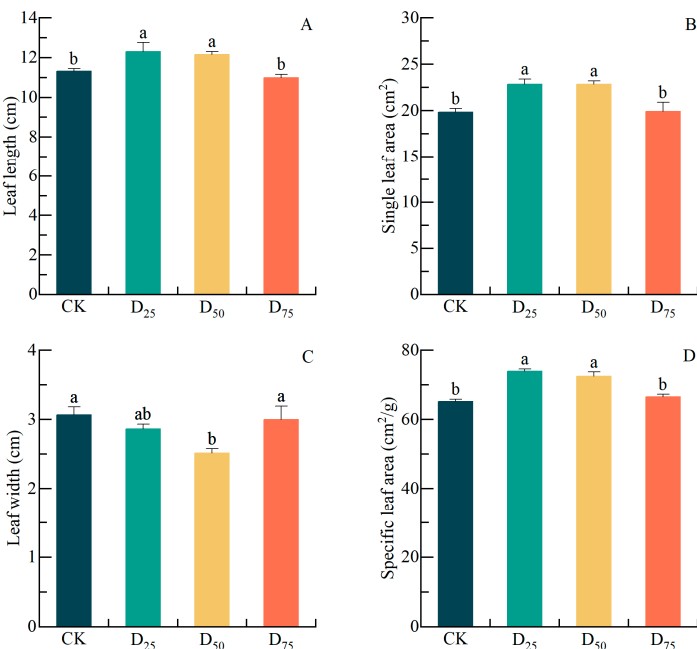

**Figure 1.** Effect of leaf damage on the leaf characters in *P. talassica × P. euphratica*. Different letters represent statistically significant differences ($p < 0.05$). (**A**): Effects of different damage treatments on the leaf length in *P. talassica × P. euphratica*. (**B**): Effects of different damage treatments on the single leaf area in *P. talassica × P. euphratica*. (**C**): Effects of different damage treatments on the leaf width in *P. talassica × P. euphratica*. (**D**): Effects of different damage treatments on the specific leaf area in *P. talassica × P. euphratica*.

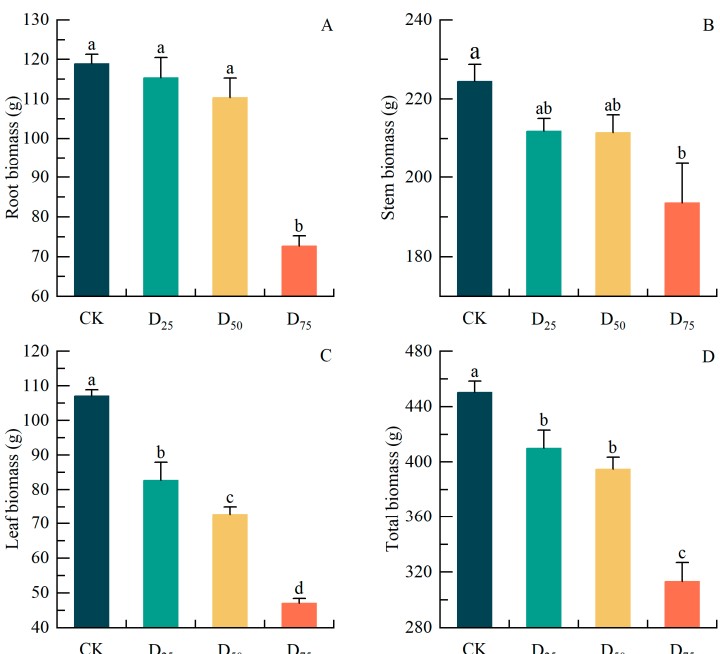

**Figure 2.** Effect of leaf damage on *P. talassica × P. euphratica* biomass. Different letters represent statistically significant differences ($p < 0.05$). (**A**): Effects of different damage treatments on the root biomass in *P. talassica × P. euphratica*. (**B**): Effects of different damage treatments on the stem biomass in *P. talassica × P. euphratica*. (**C**): Effects of different damage treatments on the leaf biomass in *P. talassica × P. euphratica*. (**D**): Effects of different damage treatments on the total biomass in *P. talassica × P. euphratica*.

### 3.4. Damage Treatment Effects on R/S, QI and CI

The R/S, QI, and CI are shown in Figure 3. Compared with CK, R/S was significantly reduced by 14.08% in the $D_{75}$ treatment, which has slightly increased under the $D_{25}$ and $D_{50}$ treatments; however, there was no statistically significant difference (Figure 3A). The QI was significantly lower after damage treatments compared with CK, by 12.93%, 11.91%, and 36.26%, respectively. However, there was no significant difference between the $D_{25}$ and $D_{50}$ treatments (Figure 3B). The CI decreased as damage intensity increased, which was significantly reduced in the $D_{75}$ treatment compared with the $D_{25}$ and $D_{50}$ treatments. However, the CI of all treatments was less than 1, which indicated undercompensated growth (Figure 3C).

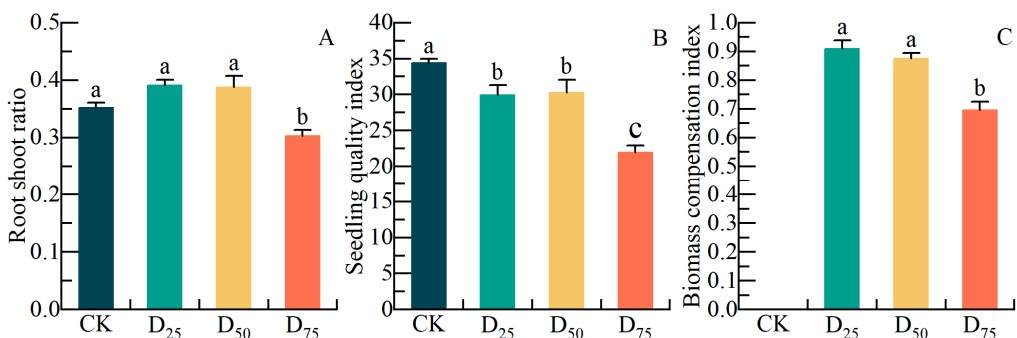

**Figure 3.** Effect of leaf damage on root-to-shoot ratio, seedling quality index, and biomass compensation index in *P. talassica* × *P. euphratica*. Different letters represent statistically significant differences ($p < 0.05$). (**A**): Effects of different damage treatments on the root shoot ratio in *P. talassica* × *P. euphratica*. (**B**): Effects of different damage treatments on the seedling quality index in *P. talassica* × *P. euphratica*. (**C**): Effects of different damage treatments on the biomass compensation index in *P. talassica* × *P. euphratica*.

### 3.5. Damage Treatment Effects on Chlorophyll Contents

Changes in the chlorophyll content are shown in Table 3. After 5 d, the Chl a, Chl b, and Total Chl contents under the $D_{25}$ treatment were significantly greater than under other damage treatment levels; however, there was no significant difference compared with CK. At 10 d and 30 d, the Chl a, Chl b, and Total Chl contents in the $D_{50}$ treatment were significantly greater than in CK, whereas there was no significant difference between other damage treatment levels and CK. After 60 d, the Chl a, Chl b, and Total Chl contents significantly increased under the $D_{25}$ and $D_{75}$ treatments compared with CK, whereas the difference between the $D_{50}$ treatment and CK was not significant. As the treatment time increased, the changes in the Chl a, Chl b, and Total Chl contents compared with CK were as follows: $D_{25}$, high-flat-high; $D_{50}$, flat-high-flat; and $D_{75}$, low-flat-high.

### 3.6. Damage Treatment Effects on Photosynthetic Parameters

After 30 d, compared with CK, under the $D_{50}$ and $D_{75}$ treatments, the *Pn* significantly increased by 31.22% and 28.65%, respectively (Figure 4A). The *Gs* significantly increased (Figure 4B). The change in *Ci*, compared with CK, showed a trend of first increasing and then decreasing as the damage intensity increased. The smallest change occurred under the $D_{50}$ treatment (Figure 4C). Compared with CK, the *E* significantly increased only under the $D_{25}$ treatment; however, there was no significant difference in the other two treatments (Figure 4D). Under the $D_{50}$ and $D_{75}$ treatments, the *WUE* significantly increased compared with the CK, with an increases in 14.85% and 15.16%, respectively. The $D_{25}$ treatment did not show significant differences (Figure 4E). The changes in *Ls* were opposite those seen for *Ci*, with the maximum mean value occurring under the $D_{50}$ treatment (Figure 4F). Compared with CK, under the $D_{50}$ treatment, the *Ci* decreased and the *Ls* increased. However, when the damage intensity continued to increase to 75%, the *Ci* increased and the *Ls* decreased.

**Table 3.** Effect of leaf damage on *P. talassica* × *P. euphratica* leaf chlorophyll content.

| Days After Treatment (d) | Treament | Chl a Content (mg/g) | Chl b Content (mg/g) | Total Chl Content (mg/g) |
|---|---|---|---|---|
| 5 d | CK | 2.118 ± 0.006 ab | 0.795 ± 0.026 ab | 2.913 ± 0.021 ab |
| | D25 | 2.242 ± 0.073 a | 0.852 ± 0.039 a | 3.094 ± 0.084 a |
| | D50 | 2.045 ± 0.075 b | 0.738 ± 0.039 bc | 2.783 ± 0.114 b |
| | D75 | 1.843 ± 0.013 c | 0.678 ± 0.007 c | 2.520 ± 0.013 c |
| 10 d | CK | 2.040 ± 0.052 bc | 0.743 ± 0.015 ab | 2.783 ± 0.067 bc |
| | D25 | 1.972 ± 0.017 c | 0.703 ± 0.024 b | 2.675 ± 0.037 c |
| | D50 | 2.145 ± 0.010 a | 0.789 ± 0.004 a | 2.934 ± 0.006 a |
| | D75 | 2.122 ± 0.023 ab | 0.758 ± 0.012 a | 2.879 ± 0.033 ab |
| 30 d | CK | 1.431 ± 0.002 bc | 0.519 ± 0.009 ab | 1.950 ± 0.010 bc |
| | D25 | 1.509 ± 0.028 ab | 0.556 ± 0.015 a | 2.065 ± 0.043 ab |
| | D50 | 1.553 ± 0.011 a | 0.559 ± 0.010 a | 2.112 ± 0.010 a |
| | D75 | 1.367 ± 0.051 c | 0.497 ± 0.024 b | 1.863 ± 0.075 c |
| 60 d | CK | 1.555 ± 0.081 b | 0.546 ± 0.035 b | 2.101 ± 0.115 b |
| | D25 | 1.899 ± 0.055 a | 0.708 ± 0.011 a | 2.607 ± 0.065 a |
| | D50 | 1.540 ± 0.041 b | 0.544 ± 0.012 b | 2.085 ± 0.053 b |
| | D75 | 1.756 ± 0.008 a | 0.698 ± 0.002 a | 2.454 ± 0.007 a |

Note: The data in the table represent Mean ±standard errors, and different lowercase letters indicate significant differences ($p < 0.05$).

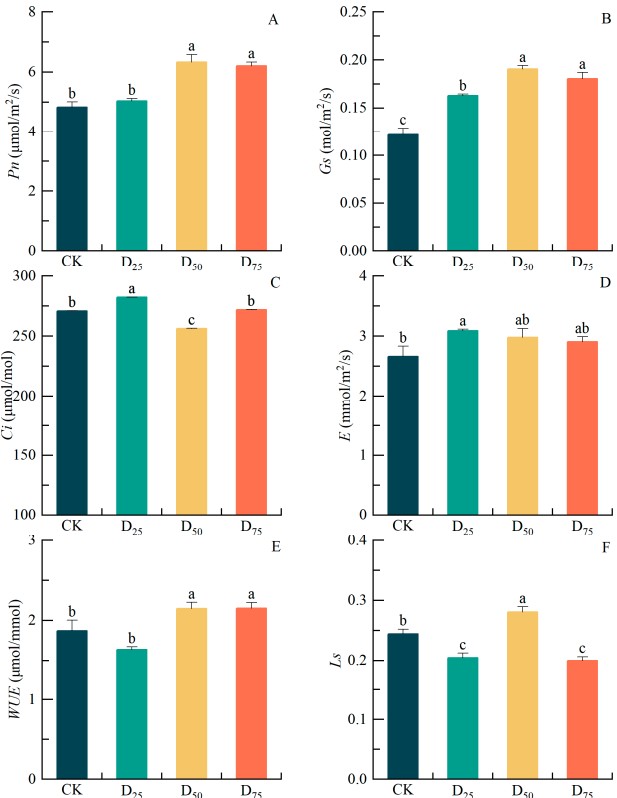

**Figure 4.** Effect of leaf damage on photosynthetic parameters of *P. talassica* × *P. euphratica*. Different letters represent statistically significant differences ($p < 0.05$). (**A**): Effects of different damage treatments on the e net photosynthetic rate in *P. talassica* × *P. euphratica*. (**B**): Effects of different damage treatments on the stomatal conductance in *P. talassica* × *P. euphratica*. (**C**): Effects of different damage treatments on the intercellular $CO_2$ concentration in *P. talassica* × *P. euphratica*. (**D**): Effects of different damage treatments on the transpiration rate in *P. talassica* × *P. euphratica*. (**E**): Effects of different damage treatments on the water-use efficiency in *P. talassica* × *P. euphratica*. (**F**): Effects of different damage treatments on the stomatal limitation values in *P. talassica* × *P. euphratica*.

### 3.7. Damage Treatment Effects on Stomatal Characteristics

The stomatal numbers and the proportions of opened stomata varied after 30 d of leaf damage (Figure 5). The stomatal numbers were relatively lower in the treatment group compared with CK, and this was essentially consistent at different damage intensities (Figure 5A–D). The proportions of opened stomata were significantly greater, and the stomatal openings were larger in the $D_{25}$ and $D_{50}$ treatments than in the CK and $D_{75}$ treatments (Figure 5E–H).

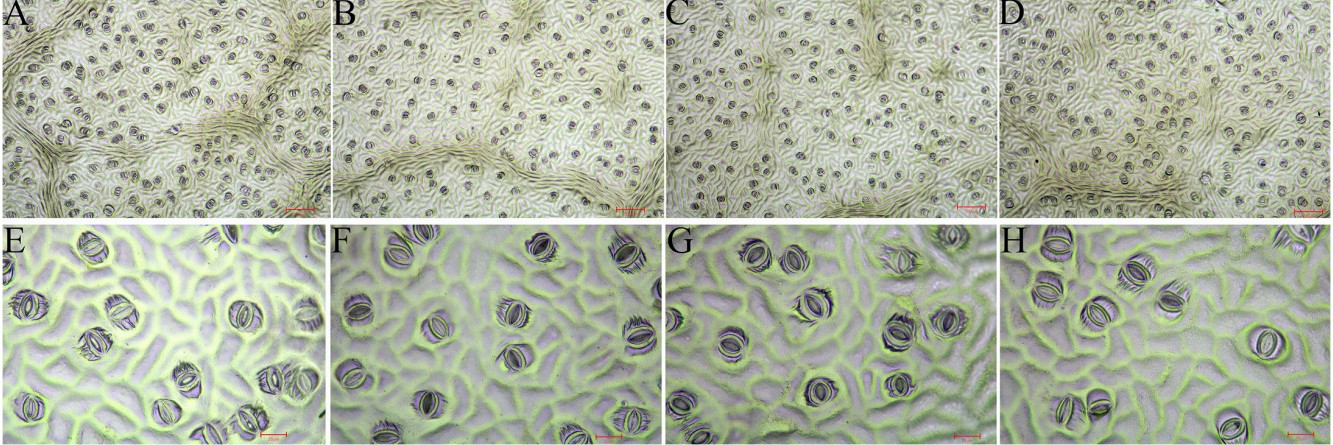

**Figure 5.** Effect of leaf damage on the stomatal morphological characteristics of *P. talassica* $\times$ *P. euphratica* leaves. (**A–H**) Micortructures of leaf stomata. (**A**): CK ($\times$10); (**B**): $D_{25}$ ($\times$10); (**C**): $D_{50}$ ($\times$10); (**D**): $D_{75}$ ($\times$10); (**E**): CK ($\times$40); (**F**): $D_{25}$ ($\times$40); (**G**): $D_{50}$ ($\times$40); (**H**): $D_{75}$ ($\times$40).

The stomatal characteristics of *P. talassica* $\times$ *P. euphratica* leaves after damage treatments are presented in Table 4. Leaf damage significantly increased the stomatal density; however, there were no significant differences among the different treatments. On the contrary, leaf damage significantly increased the stomatal width, area, and aperture of *P. talassica* $\times$ *P. euphratica*. Among them, under the $D_{50}$ treatment, the stomatal width, area, and aperture significantly increased compared with CK, increasing by 8.59%, 8.40%, and 23.27%, respectively, whereas the stomatal length significantly decreased by 4.53% compared with CK. The proportions of opened stomata showed significant differences in response to leaf damage, showing a trend of first increasing and then decreasing. The variation trend of the stomatal shape index was similar to the former, with the smallest, which were close to circular, occurring under the $D_{50}$ treatment; however, there were no significant differences in the stomatal area indexes.

**Table 4.** Effect of leaf damage on the stomatal morphological parameters in *P. talassica* $\times$ *P. euphratica* leaves.

| Treatment | Stomatal Length (µm) | Stomatal Width (µm) | Stomatal Area (µm²) | Stomatal Density (number/mm²) | Stomatal Aperture (µm) | Proportion of Opened Stomata (%) | Stomatal Shape Index | Stomatal Area Index |
|---|---|---|---|---|---|---|---|---|
| CK | 26.755 ± 0.197 a | 14.273 ± 0.204 b | 264.358 ± 3.051 b | 171.200 ± 2.133 a | 7.684 ± 0.219 c | 0.585 ± 0.112 c | 1.173 ± 0.007 a | 0.045 ± 0.017 a |
| $D_{25}$ | 26.495 ± 0.363 a | 14.312 ± 0.358 b | 266.592 ± 3.920 b | 154.000 ± 2.813 b | 8.351 ± 0.139 b | 0.631 ± 0.112 b | 1.144 ± 0.083 b | 0.041 ± 0.010 a |
| $D_{50}$ | 25.544 ± 0.348 b | 15.499 ± 0.367 a | 286.558 ± 5.597 a | 157.200 ± 1.890 b | 9.472 ± 0.368 a | 0.650 ± 0.192 a | 1.103 ± 0.088 c | 0.045 ± 0.017 a |
| $D_{75}$ | 27.314 ± 0.310 a | 15.215 ± 0.117 a | 281.428 ± 4.735 a | 151.200 ± 2.048 b | 8.536 ± 0.289 b | 0.500 ± 0.017 d | 1.132 ± 0.073 b | 0.042 ± 0.013 a |

Note: The data in the table represent Mean ±standard errors, and different lowercase letters indicate significant differences ($p < 0.05$).

### 3.8. Damage Treatment Effects on Anatomical Structure of Stems and Leaves

As shown in Figure 6, the upper and lower epidermis of *P. talassica* $\times$ *P. euphratica* leaves are all tightly arranged monolayer cells, and the mesophyll differentiates into

palisade and sponge tissues. The palisade tissue is arranged in two to three layers of long columnar cells, which are relatively dense. The sponge tissue cells have irregular shapes and loose arrangements. The vascular bundle of the main vein is composed of xylem, phloem, and vascular cambium. There is sclerenchyma outside the vascular bundle, and there are dark ergastic substances in the phloem and abaxial parenchymal cells to enhance the leaf water-holding capacity. The anatomical leaf diagram shows that under the $D_{75}$ treatment, the amount of dark ergastic substance decreased, and some cells ruptured. The arrangement of the palisade tissue was loose, and the arrangement of the sponge tissue was chaotic, with large gaps in the middle (Figure 6D,H).

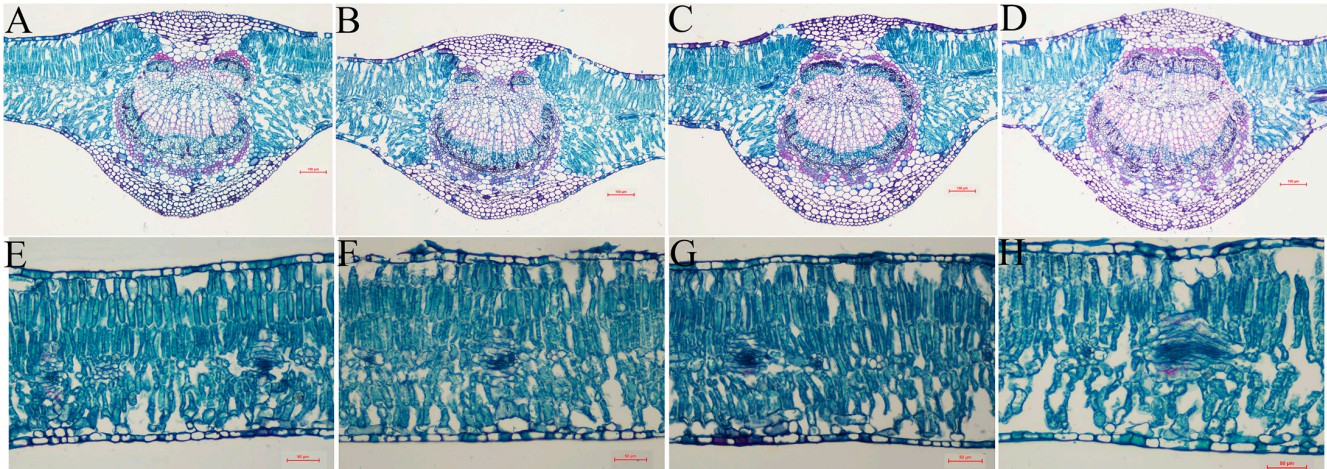

**Figure 6.** Effect of leaf damage on the anatomical structure and main veins in *P. talassica* × *P. euphratica* leaves. (**A–D**) Microstructures of leaf vascular bundles. (**A**): CK (×10); (**B**): $D_{25}$ (×10); (**C**): $D_{50}$ (×10); (**D**): $D_{75}$ (×10). (**E–H**) Microstructures of mesophyll. (**E**): CK (×20); (**F**): $D_{25}$ (×20); (**G**): $D_{50}$ (×20); (**H**): $D_{75}$ (×20).

Table 5 demonstrated the effects of different damage intensities on anatomical indicators of *P. talassica* × *P. euphratica* leaves. Under the $D_{25}$ treatment, there were no significant changes in the anatomical structural parameters. Under the $D_{50}$ treatment, the upper epidermis thickness, leaf thickness, palisade tissue thickness, palisade/spongy tightness, and leaf structure tightness were significantly higher than in CK, whereas the sponge tissue thickness did not show significant changes. Under the $D_{75}$ treatment, the sponge tissue thickness was significantly reduced compared with CK, whereas the other anatomical structural parameters showed the opposite trend.

**Table 5.** Effect of leaf damage on the anatomical structures in *P. talassica* × *P. euphratica* leaves.

| Treatment | Upper Epidermis Thickness (μm) | Leaf Thickness (μm) | Palisade Tissue Thickness (μm) | Spongy Tissue Thickness (μm) | Palisade/Spongy | Leaf Structure Tightness |
|---|---|---|---|---|---|---|
| CK | 12.782 ± 0.115 c | 294.944 ± 1.417 c | 135.483 ± 0.707 c | 125.255 ± 0.653 a | 1.082 ± 0.009 c | 0.459 ± 0.002 c |
| $D_{25}$ | 12.958 ± 0.100 c | 297.447 ± 0.530 bc | 137.116 ± 0.579 c | 124.994 ± 0.364 a | 1.097 ± 0.006 c | 0.461 ± 0.002 bc |
| $D_{50}$ | 13.714 ± 0.133 b | 299.563 ± 1.144 b | 140.512 ± 0.775 b | 125.517 ± 0.628 a | 1.120 ± 0.008 b | 0.469 ± 0.004 b |
| $D_{75}$ | 14.081 ± 0.125 a | 310.119 ± 1.379 a | 151.972 ± 0.897 a | 120.861 ± 0.305 b | 1.257 ± 0.007 a | 0.490 ± 0.003 a |

Note: The data in the table represent Mean ±standard errors, and different lowercase letters indicate significant differences ($p < 0.05$).

The stem cross sections were nearly circular in *P. talassica* × *P. euphratica*, and they are composed of the epidermis, cortex, vascular bundle, and pith. The epidermis is a tightly arranged single layer of cells. The cortex is composed of collenchyma and cortical parenchyma cells. The phloem is closely connected to the parenchyma of the cortex and is narrower than the xylem. In addition, phloem fibers, phloem parenchyma cells, and other structures can be observed. The vessels of the xylem are radially distributed, and wood

rays, wood parenchyma cells, and vessels can be observed. The pith is located in the center of the stem and consists of large parenchymal cells, which have a storage function. Under the $D_{50}$ treatment, the anatomical diagram of the stem segments shows that the thicknesses of the phloem and xylem increased, as did the number of xylem vessels (Figure 7C,G).

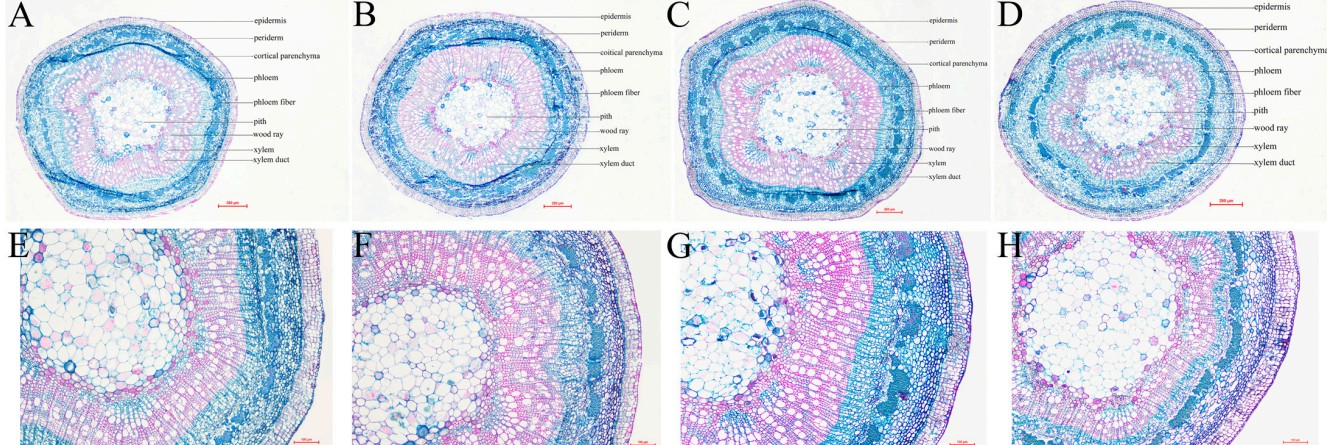

**Figure 7.** Effects of leaf damage on the anatomical structures in *P. talassica* $\times$ *P. euphratica* stems. (**A**): CK ($\times$4); (**B**): $D_{25}$ ($\times$4); (**C**): $D_{50}$ ($\times$4); (**D**): $D_{75}$ ($\times$4); (**E**): CK ($\times$10); (**F**): $D_{25}$ ($\times$10); (**G**): $D_{50}$ ($\times$10); (**H**): $D_{75}$ ($\times$20).

Changes in the anatomical structural parameters of stem segments are provided in Table 6. As the damage intensity increased, anatomical indexes such as the epidermis thickness, phloem thickness, xylem thickness, and pith diameter increased first and then decreased. Compared with CK, these parameters significantly increased by 7.74%, 14.15%, 51.67%, and 24.56%, respectively, under the $D_{50}$ treatment, whereas the cortical thickness did not change significantly under the damage treatments.

**Table 6.** Effect of leaf damage on the anatomical structures in *P. talassica* $\times$ *P. euphratica* stems.

| Treatment | Epidermi Thickness (µm) | Cortica Thickness (µm) | Phloem Thickness (µm) | Xylem Thickness (µm) | Pith Diameter (µm) |
|---|---|---|---|---|---|
| CK | 10.703 ± 0.189 b | 115.900 ± 3.051 a | 167.899 ± 1.959 b | 227.513 ± 2.012 b | 747.603 ± 4.088 b |
| $D_{25}$ | 10.898 ± 0.239 ab | 119.139 ± 1.005 a | 166.229 ± 1.520 b | 229.445 ± 2.751 b | 743.546 ± 8.542 b |
| $D_{50}$ | 11.531 ± 0.167 a | 119.364 ± 1.169 a | 191.663 ± 2.656 a | 345.070 ± 3.861 a | 931.236 ± 2.349 a |
| $D_{75}$ | 11.335 ± 0.239 ab | 114.504 ± 1.543 a | 148.681 ± 2.189 c | 226.013 ± 1.698 b | 756.657 ± 8.045 b |

Note: The data in the table represent Mean ±standard errors, and different lowercase letters indicate significant differences ($p < 0.05$).

### 3.9. Damage Treatment Effects on Enzyme Activities

The effects of different damage intensities on the defense enzyme activities of *P. talassica* $\times$ *P. euphratica* leaves are shown in Figure 8. Different damage intensities induced CAT activity (Figure 8A). After 1 d of the $D_{25}$ treatment, CAT levels were significantly higher than in CK, reaching a maximum at 4 d and then decreasing to no significant difference from CK by 7 d. At 1–7 d after beginning the $D_{50}$ treatment, there was a significant difference in CAT activity compared with CK, and it showed a trend of first increasing and then decreasing. However, by 10 d, there was no significant difference in CK. The CAT activity induced by the $D_{75}$ treatment reached its maximum mean value at 1 d, significantly increasing by 81.85% compared with CK, and it was significantly higher than values under the $D_{25}$ and $D_{50}$ treatments. It then rapidly decreased to no significant difference from CK by 7 d.

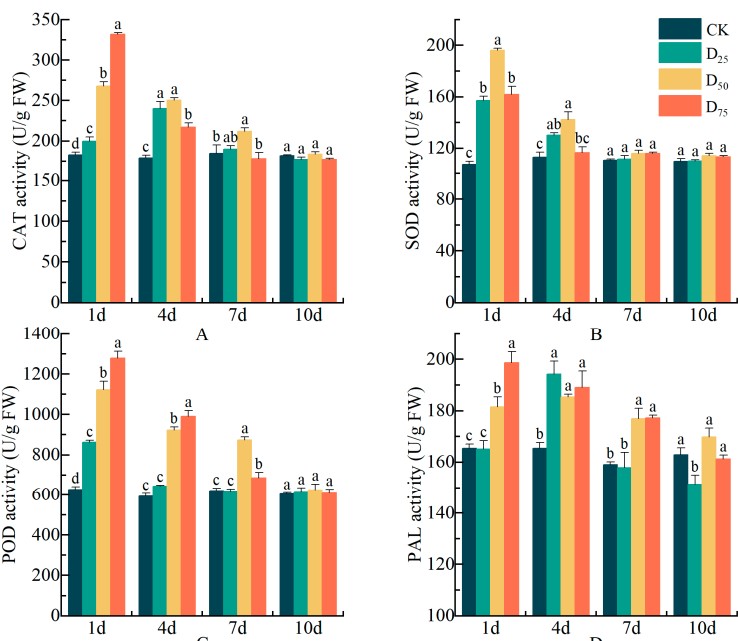

**Figure 8.** Effect of leaf damage on defensive enzyme activities in *P. talassica* × *P. euphratica*. Different letters represent statistically significant differences ($p < 0.05$). (**A**): CAT; (**B**): SOD; (**C**): POD; (**D**): PAL.

As the damage intensity increased, the SOD activity showed a trend of first increasing and then decreasing (Figure 8B). After the different damage treatments, the SOD change trends over treatment time were similar. After 1 d of damage treatment, the SOD activity rapidly increased, and the difference from CK reached a significant level. Subsequently, the SOD levels decreased to no significant difference from CK at 7 d. The SOD activity induced by the $D_{50}$ treatment reached its maximum mean value, significantly increasing by 82.73% compared with CK by 1 d. There was no significant difference in SOD activity between the $D_{25}$ and $D_{75}$ treatments.

The changes in POD activities were similar to the CAT trend (Figure 8C). Under the $D_{25}$ treatment, the POD activity was highest at 1 d, being significantly higher than CK, and then decreased to no significant difference from CK by 4 d. The POD remained high from 1 d to 7 d after the $D_{50}$ treatment, but then it decreased to no significant difference from CK by 10 d. For the $D_{75}$ treatment, the most significant change was at 1 d, when the POD activity significantly increased by 1.04-fold compared with CK. Subsequently, it rapidly decreased to no significant difference from CK by 10 d.

The PAL activity of *P. talassica* × *P. euphratica* leaves significantly changed over time under different damage intensities (Figure 8D). The PAL activity induced by the $D_{25}$ treatment reached its maximum mean value at 4 d and then rapidly decreased to be significantly lower than in CK by 10 d. The PAL activities induced by the $D_{50}$ and $D_{75}$ treatments remained high from 1 d to 7 d, being significantly different from CK, and then decreased to no significant difference from CK by 10 d.

The activities of the four defense enzymes showed trends of first increasing and then decreasing as damage time increased. CAT, POD, and PAL activities quickly reached their maximum mean values after 1 d of the $D_{75}$ treatment, and under the $D_{50}$ treatment, the high enzyme activity levels lasted longer than in other treatment groups.

## 4. Discussion

### 4.1. Effects of Leaf Damage on Growth and Biomass of P. talassica × P. euphratica

The regenerative capacity of woody plants is an important component in maintaining vegetative diversity in forest ecosystems [26]. Many woody plants initiate regulatory mechanisms to produce compensatory growth after suffering mechanical damage that results in leaf damage and loss [27]. The compensatory growth ability of plants is related

to factors such as species, damage intensity, and recovery time [28,29]. Compensatory growth is a positive response to plant damage and can be divided into three types: growth promotion after damage is termed "super-compensatory growth", biomass change after damage is insignificant and termed "iso-compensatory growth", and growth inhibition after damage is termed "under-compensatory growth" [17]. Growth indicators are the most intuitive gauges of plant growth and reveal the adaptive capacities of plants. Biomass accumulation reflects the carbon sequestration capacity of a plant. To compensate for the loss of above-ground biomass, plants can adopt various strategies to restore growth. For example, they can allocate more resources to the growth of above-ground organs or possibly increase SLA to improve photosynthetic use efficiency [30,31].

The effects of leaf loss on growth are not uniform across plants. For example, willows of different genotypes undergo different growth trends after defoliation, with some genotypes showing significant changes after 75% loss [32]. In *Gmelina arborea,* a 50% leaf loss rate significantly reduces plant height and biomass, but it does not significantly affect ground diameter [33]. In *Quercus acutissima*, a 50% damage treatment resulted in a significant decrease in plant height, ground diameter, and biomass, but it significantly increased the specific leaf area [34]. The results showed that the reductions in different growth parameters became greater with increasing damage intensity. It is worth noting that leaf damage has a significant impact on crown width growth. Except for the 30 d after $D_{25}$ treatments with no significant difference compared to CK, other treatments were significantly reduced. It was speculated that the loss of leaves near the terminal buds may have slowed down the growth of the branches. Under the $D_{75}$ treatment, the growth index and biomass of *P. talassica* $\times$ *P. euphratica* were significantly reduced, crown growth was inhibited during the first part of the treatment (30 d), and the R/S and QI values were significantly reduced. However, under the $D_{25}$ and $D_{50}$ treatments, there were no effects on growth indexes such as root biomass and stem biomass, but there was an increase in leaf length, leaf area, and SLA. In summary, *P. talassica* $\times$ *P. euphratica* have a certain ability to compensate for leaf damage; however, when the damage reached 75%, it inhibited plant growth, reduced the accumulation of root, stem, and leaf biomass, and reduced the carbon sequestration capacity. Thus, plants were unable to support the regeneration of above-ground parts, leading to under-compensated growth, which is consistent with the results of CI.

*4.2. Effects of Leaf Damage on Physiological Characteristics of P. talassica $\times$ P. euphratica*

Enhanced photosynthesis is a common and important compensatory mechanism following leaf damage [35], and studying changes in the photosynthetic capacities of plants following damage treatments increases our understanding of the physiological basis of plant damage responses. As early as the 1960s, it was suggested that the photosynthetic capacity of plants was enhanced after leaf loss [36]. In *Petunia nigra*, changes in the photosynthetic capacity vary according to the type of leaf damage, with the removal of leaf margins, removal of leaf tips, and perforation all significantly increasing the photosynthetic capacity; however, the removal of leaf margins is more effective in enhancing the *Pn* and *Gs* of the plant than the other two treatments [37]. In *Glycine max*, *Pn* also increases significantly after damage treatments, but *Ci* significantly decreases [38]. However, in *Quercus acutissima* seedlings with sufficient light and soil nutrients, the *Pn* significantly decreases after cotyledon damage treatments, and the chlorophyll content only decreases after severe damage [19]. In this study, the chlorophyll content of *P. talassica* $\times$ *P. euphratica* took longer to recover as the damage intensity increased. The chlorophyll content was not reduced by the $D_{25}$ treatment. It took 10 d to exceed the CK level under the $D_{50}$ treatment, but then it remained high. Under the $D_{75}$ treatment, the chlorophyll level also increased, but it returned to the CK level by 60 d. Under the $D_{50}$ and $D_{75}$ treatments, *Pn, Gs,* and *WUE* increased significantly, and a photosynthetic compensation effect was observed. We hypothesized that after leaf damage in *P. talassica* $\times$ *P. euphratica*, the new leaves increase their carbon demand, stimulating the photosynthetic capacity of the remaining leaves and contributing to the up-regulation. It may also be related to changes in leaf age and

photosynthesis-related enzyme levels [39]. In *P. talassica* × *P. euphratica* suffering from leaf damage, the light capture efficiency increased, which enhanced photosynthesis per unit area by increasing the chlorophyll content, SLA, and light transmission. The effects on photosynthesis due to the loss of leaf area were compensated for, thereby offsetting the effects on the growth and biomass of *P. talassica* × *P. euphratica*.

Stomata, important structures composed of two Guard cells, exchange water and gas with the outside world. Plants respond to adverse effects through stomatal regulation, which aids in maintaining their normal growth [40]. In this study, leaf damage significantly reduced stomatal density, possibly due to an increase in leaf area. Stomatal width, area, and aperture significantly increased under the $D_{50}$ and $D_{75}$ treatments. This may have resulted from the plants' abilities to ensure sufficient water and air exchange for normal photosynthesis, thereby alleviating leaf damage. However, under the $D_{75}$ treatment, the proportion of opened stomata actually decreased, likely due to excessive damage to the plant leaves, which affected the number of open stomata. Thus, plants appear to undergo plastic changes in morphology and physiology to adapt to environmental changes [41].

*4.3. Effects of Leaf Damage on Anatomical Structures of P. talassica × P. euphratica*

The leaf is an important organ for physiological metabolic activities such as photosynthesis, transpiration, and respiration, and leaves often alter their morphological and anatomical structures when habitat conditions change, thus exhibiting strong environmental sensitivity and plasticity [42]. Leaf structure is a factor that affects the photosynthetic efficiency of plants, and, to a certain extent, it can reflect the physiological adaptations of plants [43]. The higher the proportion of palisade tissue in the leaf structure, the higher the chlorophyll content, the tighter the plant tissue, and the stronger its photosynthetic performance [44]. In addition, the increase in leaf thickness and the thickness of the upper and lower epidermis reduce water transpiration and improve water retention capacity [45]. This study corroborated these findings. In this study, anatomical parameters such as palisade tissue thickness, palisade/spongy, and tightness of leaf tissue structure increased significantly under the $D_{50}$ and $D_{75}$ treatments. We hypothesized that *P. talassica* × *P. euphratica* mitigated the effects of leaf damage on the photosynthetic capacity by changing the intrinsic leaf structure, especially by increasing palisade tissue thickness. When the damage intensity reached 75%, although leaf thickness and upper epidermal thickness increased to reduce water evaporation, the palisade tissues became loosely arranged and the cell gaps became larger, while the spongy tissues' thickness decreased, became disorganized, and large gaps appeared. This indicated that the reduced carbon reserves caused by high-damage intensity may hinder water metabolism in the leaves, which is detrimental to normal leaf growth and development.

The internal structure of a stem, an important organ for transporting water and nutrients in plants, affects uptake and plant development. Leaf loss also affects the stem's ability to absorb water and reduces the resistance of the xylem to embolism [46]. These anatomical indicators, such as xylem/phloem, a well-developed vascular bundle and pith, and the size and density of vessels, are important factors to enhance the water and nutrients transport efficiency of the stem [47,48]. Leaf damage treatments did not significantly affect cortical thickness, but they significantly affected the phloem thickness, xylem thickness, and pith diameter. Under the $D_{50}$ treatment, the xylem was more developed, had more ducts, and was nearly rounded, indicating that *P. talassica* × *P. euphratica* stems had higher water transport efficiency and stronger resistance, so that they could better adapt to the damage caused by leaf loss.

*4.4. Effects of Leaf Damage on Defense Enzyme Activities of P. talassica × P. euphratica*

Induced defense mechanisms play important roles in plant self-protection. During long-term growth and development, plants tend to form defense mechanisms for self-protection to avoid or reduce insect feeding or other mechanical damage, and the synthesis of defense enzymes is an important expression of the defense system's formation [49]. The

important defense enzymes CAT, SOD, POD, and PAL are closely related to plant-induced resistance and damage tolerance [13,14,50].

Partial damage to plants can induce the expression of defense signaling substances [51], which strongly induce an increase in defense enzyme activities in a short period of time. The amount and duration of the enzymatic expression and their maximum activities are related to the damage intensity, kinds of enzymes, and species. For example, after mechanical damage to the hybrid poplar, which generated the accumulation of $H_2O_2$ content in unwounded leaves and the significant enhancement of CAT and SOD activities, the two enzymes reached their peak at different times [52]. Under the conditions of less than 25% and more than 25% of fallen leaves, there was a significant difference in CAT, SOD, and POD of *Pinus nigra*, while only CAT activity showed a significant difference in *Quercus pubescens* and *Pinus halepensis* [53]. This trend is the same as that of the enzymatic activity induced by pine caterpillar feeding. After the artificial defoliation of *Pinus cumerosae*, the POD activity significantly increased by 3 d; however, there was no significant change in the SOD activity [54]. The POD activity in the leaves of *Pinus sylvestris* L. increases briefly at 72 h after defoliation and then rapidly decreases [50]. In addition, mechanical damage can stimulate an increase in PAL activity, which induces resistance in plants [55]. Here, the different damage intensities induced an increase in defense enzyme activities in the remaining leaves, indicating that the four defense enzymes act synergistically in resisting external damage. However, the different damage intensities had different effects on the temporal changes in enzymatic activities, indicating that the various defense enzymes responded differently to the leaf damage treatments. In the $D_{50}$ and $D_{75}$ treatments, the CAT and POD activities were strongly induced, reaching maximum values at 1 d. The former induced a longer duration of CAT activity, whereas the latter resulted in a rapid decrease. A similar trend in SOD activity, but with a relatively shorter duration, was also seen. This is probably due to the large accumulation of $H_2O_2$ in damage-stimulated *P. talassica* $\times$ *P. euphratica*, which promoted the enhancement of CAT and POD activities, while SOD acted to maintain the steady-state level of reactive oxygen radicals. Thus, after damage, the activities of the defense enzymes were rapidly stimulated to remove the oxygen radicals accumulated in the leaves, inhibit membrane lipid peroxidation, and protect against external damage. Although the response of the compensatory growth capacity to different damage intensities differed, all the damage treatments activated the defense enzyme activities in the remaining leaves. Thus, we assumed that *P. talassica* $\times$ *P. euphratica* have a certain capability to perceive damage and activate defense mechanisms quickly when stimulated by external injury, and this plays an important physiological role in plant growth.

Plant adaptation to damage is a very complex process. Damage induction in *Populus* is currently poorly studied, and the order and coordinated roles of various associated substances in plants need to be determined. In this study, we showed that *P. talassica* $\times$ *P. euphratica* have a certain capability to repair damage. When the leaves are damaged, they adjust the resource allocation strategy and physiological defense capacity by increasing the chlorophyll content, improving the photosynthetic capacity, changing stem and leaf anatomical features, and increasing defense enzyme activity levels, thereby improving their damage tolerance and adaptability.

## 5. Conclusions

This study analyzed the effects of leaf damage on *P. talassica* $\times$ *P. euphratica* to investigate compensatory growth and physiological defenses in terms of growth and development, photosynthetic capacity, anatomical structure, and defense enzyme activities. Minor leaf damage ($D_{25}$) did not affect plant growth, although it increased defense enzyme activities in the short term. The reductions in photosynthetic area and carbon assimilation capacity under severe damage ($D_{75}$) were the main causes of the energy imbalance and under-compensated growth. Under the $D_{50}$ treatment, the plant also rapidly activated the antioxidative defense system and improved its defense capacity, which played important physiological protective roles in compensatory growth and strong damage tolerance. There

are limited studies on the compensatory growth of woody plants due to leaf damage. The processes involved in plant responses to leaf damage are very complex, and there are multiple habitat variables that interact with leaf damage to affect plant growth. Consequently, it is difficult to accurately assess the threshold of the degree of leaf damage on plant growth. Thus, the survival risk caused by leaf damage during the initial establishment of plantation forests should be taken into consideration.

**Author Contributions:** Conceptualization, Z.-J.H.; methodology, Z.-J.H. and M.-X.S.; software and data curation, M.-X.S.; investigation and validation, M.-X.S., Y.L., Z.Z. and J.-J.W.; resources, Z.-J.H.; formal analysis, M.-X.S. and Z.-J.H.; visualization, M.-X.S. and Z.Z.; writing—original draft preparation, M.-X.S.; writing—review and editing, Z.-J.H. and M.-X.S.; supervision, Z.-J.H.; project administration, Z.-J.H. and M.-X.S.; funding acquisition, Z.-J.H. All authors have read and agreed to the published version of the manuscript.

**Funding:** This research was funded by the Selection and Cultivation Project of "Talents of Xinjiang Production and Construction Corps" (Funding number: 380000358 and 38000020924, Funder: Xinjiang Production and Construction Corps). This work was also funded by the Open Project of the Xinjiang Production and Construction Corps Key Laboratory of Protection and Utilization of Biological Resources in Tarim Basin (Funding number: BRZD1902, Funder: Xinjiang Production and Construction Corps Key Laboratory).

**Data Availability Statement:** The original contributions presented in this study are included in the article. Further inquiries can be directed to the corresponding author.

**Acknowledgments:** Thank you to De-Qiang Lu of Lumao Shengyuan Agricultural Development Co., Ltd. (Alar, China) for providing assistance with *P. talassica* × *P. euphratica* seedlings and field experimental sites.

**Conflicts of Interest:** The authors declare that there is no conflict of interest regarding the publication of this paper.

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
