# Peer review of "Compensatory Growth and Physiological Protective Mechanisms of Populus talassica Kom. × Populus euphratica Oliv. in Response to Leaf Damage"

_forests, doi:10.3390/f14091713_

Round 1
Reviewer 1 Report
Dear Authors,
It is a well-structures, well-written manuscript about a high quality experiment.
I have just minor suggestions:
Please explain me figure 3A, how could the R/S decrease with D75 considering that you artificially reduced the AB in that treatment?
I suggest to modify figures 5, 6 and mostly figure 7, because of the low visibility and understanding due to the size and quality.
Kind regards
English is fine, minor spelling mistakes and typos are present, but the manuscript provides clear understanding.
Author Response
Response to Reviewer 1 Comments:
Point 1: Please explain me figure 3A, how could the R/S decrease with D75 considering that you artificially reduced the AB in that treatment?
Response 1: Thank you very much for your valuable comments. Plants are divided into two parts, roots and crowns, with root growth limited by the rate of photosynthetic carbon supply to the crown, and crown growth limited by the rate of nutrient and water uptake by the roots. Plants can obtain resources such as light, nutrients, and water through the allocation of photosynthetic products to achieve maximum growth rates. In this study, although the biomass of the aboveground part was reduced, with the increase of time, due to the significant reduction of photosynthetic organs, on the one hand, there were fewer newly formed photosynthetic products. At the same time, to stimulate the growth of new leaves, the consumption and storage of photosynthetic products increased, weakening the supply of water and nutrients to the root system. On the other hand, the leaf buds and branches left behind received a corresponding increase in the supply of water and nutrients from the root system, so that the aboveground part of the growth was better than the underground part. In summary, although leaf biomass decreases, the newly generated biomass in the aboveground part is greater than that in the underground part, resulting in a decrease in the root to shoot ratio.
Point 2: I suggest to modify figures 5, 6 and mostly figure 7, because of the low visibility and understanding due to the size and quality.
Response 2: Thank you very much for your valuable comments. We have modified the clarity of Figures 5, 6, and 7.
We acknowledge your and all the reviewer’s comments and suggestions very much, which are valuable in improving the quality of our manuscript.
Best wishes,
Meng Xu Su, Zhan Jiang Han

Reviewer 2 Report
Review Report_ijms-2476729-peer-review-v1
The manuscript ‘Compensatory growth and physiological protective mechanisms of Popular talassica × Popular euphratica in response to leaf damage’ Authors by Su et al. is well-written. However, I have a few suggestions which could improve the manuscript.
Title: Please revise ‘mechnisms’ as ‘mechanisms’.
Please use the correct nomenclature of Popular talassica Kom. And Popular euphratica Oliv.
Abstract: Please re-orient the abstract. Please write the results directly instead of numbering them. Please show the quantifiable data in the abstract.
Introduction:
Line 68: ‘southern Xinjiang’, please mention the country ‘southern Xinjiang, China’.
Line 78-80: Please re-write the aim of the study without numbering them.
Materials and Methods:
Line 85-87: Please re-write the sentence. Please write the location (Place, province, Country) along with the latitude, longitude, and altitude, as mentioned followed by the season of the study.
Line 88: Whether one season of study (May-August, 2022) is enough?? Please justify.
Line 94-96: Please remove the dot (mg·kg−1) in the unit, throughout the text.
Line 142: Why the Photosynthetic parameters were taken at 9:00–12:00 AM?? The peak Pn occurs at 11 am-1 pm. Please justify.
Line 142-150: The abbreviations Pn, Tr, Ci, Gs, Ca etc may be written in italics. The transpiration rate may be denoted as E instead of Tr.
Line 161: ‘ImageJ analysis software’, please cite a reference.
Line 205-210: Please mention the treatment and replication details, and the statistics design followed for the experiment, in the statistical analysis section.
Results and Discussion: Well written.
Line 532: CAT SOD POD. Use comma CAT, SOD, and POD.
Please try to restrict the references to 40-50, which is ideal for a manuscript.
I suggest the manuscript may be checked thoroughly for language, grammar, and punctuation.
Please cross-check the references cited properly as per the journal standard.
This is a well-structured manuscript and may be accepted with minor corrections.
Good luck with the revision.
I suggest the manuscript may be checked thoroughly for language, grammar, and punctuation.
Author Response
Response to Reviewer 2 Comments:
Title
Point 1: Please revise ‘mechnisms’ as ‘mechanisms’.
Response 1: Thank you very much for your valuable comments. We have revised the word. We are very sorry for the negligence, and have checked the manuscript.
Point 2: Please use the correct nomenclature of Popular talassica Kom. And Popular euphratica Oliv.
Response 2: Thank you very much for your valuable comments. We have revised the section.
Abstract
Point 1: Please re-orient the abstract. Please write the results directly instead of numbering them. Please show the quantifiable data in the abstract.
Response 1: Thank you very much for your valuable comments. I have revised the “Abstract” section.
Introduction
Point 1: Line 68: ‘southern Xinjiang’, please mention the country ‘southern Xinjiang, China’.
Response 1: Thank you very much for your valuable comments. We have revised the section.
Point 2: Line 78-80: Please re-write the aim of the study without numbering them.
Response 2: Thank you very much for your valuable comments. We have revised the section.
Materials and Methods
Point 1: Line 85-87: Please re-write the sentence. Please write the location (Place, province, Country) along with the latitude, longitude, and altitude, as mentioned followed by the season of the study.
Response 1: Thank you very much for your valuable comments. We have revised the section.
Point 2: Line 88: Whether one season of study (May-August, 2022) is enough?? Please justify.
Response 2: Thank you very much for your valuable comments. Winter in Xinjiang, China lasts for a long time, with low temperatures from around November to around March of the following year. Cold wave weather is often encountered in spring, and P. talassica × P. euphratica needs sufficient ground temperature to survive in cuttings. Generally, cuttings are carried out after warming up from March to April, and after September, P. talassica × P. euphratica enters the deciduous period, which limits its growth. On the one hand, it is not conducive to the conduct of experiments, on the other hand, even if experiments are conducted, the obtained data is greatly limited by weather and growth conditions, and has little reference value. Therefore, choosing to conduct this experiment from May to August can explore the purpose of the experiment.
Point 3: Line 94-96: Please remove the dot (mg·kg−1) in the unit, throughout the text.
Response 3: Thank you very much for your valuable comments. We have revised the writing method throughout the manuscript.
Point 4: Line 142: Why the Photosynthetic parameters were taken at 9:00–12:00 AM?? The peak Pn occurs at 11 am-1 pm. Please justify.
Response 4: Thank you very much for your valuable comments. In this experiment, artificial light sources were used for photosynthesis measurement to ensure relatively stable light intensity conditions. This not only eliminates the impact of light intensity on photosynthetic rate, but also facilitates the comparison of net photosynthetic rate differences between different treatments. In addition, Xinjiang time is 2 hours later than Beijing time in China, and there is a possibility of a " midday depression of photosynthesis" occurring from 12:00am-14:00pm.
Point 5: Line 142-150: The abbreviations Pn, Tr, Ci, Gs, Ca etc may be written in italics. The transpiration rate may be denoted as E instead of Tr.
Response 5: Thank you very much for your valuable comments. We have revised the writing method throughout the manuscript.
Point 6: Line 161: ‘ImageJ analysis software’, please cite a reference.
Response 6: Thank you very much for your valuable comments. We have added relevant references.
Point 7: Line 205-210: Please mention the treatment and replication details, and the statistics design followed for the experiment, in the statistical analysis section.
Response 7: Thank you very much for your valuable comments. We have revised the ”statistical analysis” section.
Results and Discussion
Point 1: Line 532: CAT SOD POD. Use comma CAT, SOD, and POD.
Response 1: Thank you very much for your valuable comments. We have revised the section and checked the manuscript for other similar errors.
Point 2: Please try to restrict the references to 40-50, which is ideal for a manuscript.
Response 2: Thank you very much for your valuable comments. We have streamlined the quantity and quality of the references. There are currently 55 references. Although the number does not match the range of suggested, these references are integral to the manuscript.
Point 3: I suggest the manuscript may be checked thoroughly for language, grammar, and punctuation.
Response 3: Thank you very much for your valuable comments. We have checked the manuscript.
Point 4: Please cross-check the references cited properly as per the journal standard.
Response 4: Thank you very much for your valuable comments. We have checked and revised the “references” section.
We acknowledge your and all the reviewer’s comments and suggestions very much, which are valuable in improving the quality of our manuscript.
Best wishes,
Meng Xu Su, Zhan Jiang Han
